# Toward characterizing cardiovascular fitness using machine learning based on unobtrusive data

**Maria Cecília Moraes Frade**[1], **Thomas Beltrame**[1,2]*, **Mariana de Oliveira Gois**[1], **Allan Pinto**[3], **Silvia Cristina Garcia de Moura Tonello**[1], **Ricardo da Silva Torres**[4], **Aparecida Maria Catai**[1]

**1** Department of Physical Therapy, Federal University of São Carlos, São Carlos, São Paulo, Brazil, **2** Samsung R&D Institute Brazil–SRBR, Campinas, São Paulo, Brazil, **3** Brazilian Synchrotron Light Laboratory (LNLS), Brazilian Center for Research in Energy and Materials (CNPEM), Campinas, São Paulo, Brazil, **4** Department of ICT and Natural Sciences, Faculty of Information Technology and Electrical Engineering, NTNU—Norwegian University of Science and Technology, Ålesund, Norway

* beltramethomas@gmail.com

**Data Availability Statement:** All relevant data are within the paper and it will be in Supporting Information files.

## Abstract

Cardiopulmonary exercise testing (CPET) is a non-invasive approach to measure the maximum oxygen uptake ($\dot{V}O_{2-max}$), which is an index to assess cardiovascular fitness (CF). However, CPET is not available to all populations and cannot be obtained continuously. Thus, wearable sensors are associated with machine learning (ML) algorithms to investigate CF. Therefore, this study aimed to predict CF by using ML algorithms using data obtained by wearable technologies. For this purpose, 43 volunteers with different levels of aerobic power, who wore a wearable device to collect unobtrusive data for 7 days, were evaluated by CPET. Eleven inputs (sex, age, weight, height, and body mass index, breathing rate, minute ventilation, total hip acceleration, walking cadence, heart rate, and tidal volume) were used to predict the $\dot{V}O_{2-max}$ by support vector regression (SVR). Afterward, the SHapley Additive exPlanations (SHAP) method was used to explain their results. SVR was able to predict the CF, and the SHAP method showed that the inputs related to hemodynamic and anthropometric domains were the most important ones to predict the CF. Therefore, we conclude that the cardiovascular fitness can be predicted by wearable technologies associated with machine learning during unsupervised activities of daily living.

## Introduction

Noncommunicable chronic diseases (NCDs) are mainly responsible for all causes of death and illness among adults aged between 35–70 years, and cardiovascular diseases are accountable for the main cause of mortality in the world [1]. There are some modifiable risk factors associated with NCDs, such as high systolic arterial pressure, high fasting plasma glucose, as well as low physical activity [2, 3].

It is known that the cardiovascular diseases and their modifiable risk factors lead to a reduction in cardiovascular fitness (CF) [4, 5]. Moreover, higher CF levels have a protective effect

**Funding:** The research was supported by Coordination for the Improvement of Higher Education Personnel grants (CAPES) 001 and (CAPES) 88887.362954/2019-00 awarded to MOG. This work was also supported by São Paulo Research Foundation (FAPESP) grant FAPESP 2016/22215-7 awarded to AMC, FAPESP 2017/09639-5 and FAPESP 2018/19016-8 awarded to TB; FAPESP 2018/22818-9 awarded to MCMF; and FAPESP 2019/16253-1 awarded to AP. The funders had no role in study design, data collection and analysis, decision to publish, or preparation of the manuscript.

**Competing interests:** The authors have declared that no competing interests exist.

against cardiovascular diseases and all-cause mortality in varied populations [4, 6, 7]. Thus, due to the considerable relevance of increasing population lifetime, the continuous measurement of CF could be considered a vital sign, and thus, it should be a priority in public health [8]; however, the definition and ways of evaluation of the CF are contradictory [9–11].

CF is commonly evaluated by measuring the maximum oxygen uptake ($\dot{V}O_{2-max}$), as the index of maximal aerobic power, obtained during cardiopulmonary exercise testing (CPET) [11–13]. The $\dot{V}O_{2-max}$ reflects the maximal capacity of the pulmonary, cardiovascular, and metabolic systems to capture, transport, and utilize oxygen, respectively, which is directly influenced by the CF [13, 14]. However, the $\dot{V}O_{2-max}$ measurement during the CPET requires trained professionals and expensive equipment [15–17], and is rarely used as a prevention tool in the general population. For this reason, the CF assessed by $\dot{V}O_{2-max}$ during CPET is not available to all populations and cannot be obtained continuously.

Therefore, considering the difficulties of performing the CPET, but given the high clinical value to assess cardiovascular fitness, new methods for continuous assessment of CF are needed. These methods could be more realistic, unobstructive, and accessible to all populations if performed outside laboratory settings, during unsupervised activities of daily living (ADL) [18]. Wearable sensors and vital signal fusion might represent a unique possibility to infer CF continuously, allowing the use of this technology in the future for pre-symptomatic detection of NCDs, especially cardiovascular diseases [6, 7].

Furthermore, there is an increasing number of studies that have combined the use of wearables and machine learning techniques for monitoring patients with NCDs, especially in the cardiorespiratory field [19, 20]. In fact, longitudinal data from wearables seem to contain enough information to predict CF of healthy volunteers during unsupervised ADL from complex machine learning algorithms [21–25].

However, despite the great potential of the combination between wearables and machine learning, there is still a lack of evidence for using these technologies to predict CF in patients with NCDs, especially in diabetes mellitus, chronic pulmonary disease, and cardiovascular diseases. Furthermore, understanding how these models, trained from machine learning algorithms, can transform vital signals into $\dot{V}O_{2-max}$ may provide complex mechanistic insights regarding the differences in CF between volunteers. Due to the complexity of the $\dot{V}O_{2-max}$ prediction algorithms based on features obtained from wearable technologies [25], the interpretability of how longitudinal vital signals are being transformed into $\dot{V}O_{2-max}$ is exceptionally low [26] because of the expected trade-off between the interpretability of a given model, and its performance to predict health outcomes [27].

Recently, explainable models have been used in medical science to better justify decision-making of the prediction models [26]. It is known that wearable sensors are useful for the continuous biological data acquisition that can be associated with machine learning techniques, such as Random Forest Regression, Neural Network and Support Vector Regression Machines to predict CF [21, 25]. Thus, understanding these models might also indicate how the human "black box" physiological systems interact with the environment, approximating the explainability of these complex algorithms to what we experience when using simpler methods, such as in linear regression models.

The SHapley Additive exPlanations (SHAP) is a valuable approach derived from the coalitional game theory, which can be used to interpret complex models built from supervised machine learning methods obtained from biological data [26, 28]. In this paper, we investigate the use of Shapley to assess the importance of features in the CF prediction problem. The main motivation for its use relies on (1) its ability to be model agnostic (i.e., method for explainability associated with any model to extract extra information about the prediction procedure

[26]. In this case, we can simply replace linear models with complex models without losing much interpretability; (2) to produce interpretations for a single data point; and (3) to produce human-friendly explanations for linear regression results when we deal with multiple regression problems. Furthermore, Shapley-based methods can produce visual interpretations in which we can easily visualize the global or local feature contributions [29].

Therefore, our main objective is to predict CF by using machine learning algorithms from data obtained by wearable technologies of volunteers with a broad spectrum of maximal aerobic power (or $\dot{V}O_{2-max}$, as an index of CF level). Afterward, an explainable artificial intelligence (AI) method will be used to investigate "how" CF can be estimated from the longitudinal signals acquired by wearables during unobtrusive experimental protocols. Our hypothesis is that machine learning algorithms can provide suitable models to predict $\dot{V}O_{2-max}$ when trained with vital signals collected by wearables, and explainable AI methods can interpret the prediction models' output of these algorithms. By doing so, we have a better understanding of how longitudinal signals during ADL are related with $\dot{V}O_{2-max}$, which has clinical implications for CF. Consequently, this study will demonstrate an innovative approach to predicting the onset of NCDs in future studies by continuously evaluating $\dot{V}O_{2-max}$, in addition to explaining the differences in CF among volunteers through explainable methods.

## Materials and methods

### Study design

This longitudinal study was approved by the Federal University of Sao Carlos Ethics Committee (CAAE: 80459817.5.1001.5504), and it was conducted in the Cardiovascular Physiotherapy Laboratory at the Federal University of Sao Carlos (UFSCar). All procedures followed the Helsinki declaration, and all volunteers signed a free and informed consent form in accordance with Resolution 466/2012 of the National Health Council.

The inclusion criteria were both sexes, ages of 18–80 years, and different levels of aerobic power (including apparently healthy volunteers, with risk factors to develop NCDs, or with type 2 diabetes mellitus, chronic obstructive pulmonary disease, or coronary arterial disease). All volunteer clinical conditions were validated from a medical diagnosis. Volunteers were excluded by orthopedic or neurological limitations; associated uncontrolled heart diseases, abnormality in the resting or exercising ECG response (infra-uneven ST-segment > 2 mm, unsustainable atrial tachycardia, atrial fibrillation or atrioventricular blocks, ventricular or supraventricular arrhythmias) that would prevent them from following the proposed protocol. Volunteers that wore the t-shirts for less than 5 days or less than 6 hours per day were also excluded. For the volunteers with NCDs, the pulse saturation ($SpO_2$) was verified at resting by pulse oximeter (sense 10, ALFAMED, Lagoa do Sino, Brazil) for safety. The experimental protocol comprised three main steps.

### Step one

During the first laboratory visit, volunteers were questioned about their health, lifestyle habits, exercise practice, and disease conditions (if present). Afterwards, they performed a physical evaluation comprising measurements such as weight, height, thorax, and abdominal circumference, and resting vital signals (heart rate, breathing rate, and arterial pressure). Finally, the researcher explained about using the wearable device, including how to wear it properly, remove and wash it (smart shirt will be further explained in the text in the future) and how to charge the battery.

## Step two

On the second day in the laboratory, volunteers performed the CPET on a cycle ergometer (Quinton Corival® 400, Seattle, USA) with a ramp-type protocol to assess the CF. The power increment was calculated using the formula described by Wasserman, considering height, age, and sex [14]. The test consisted of: (1) five minutes at rest, (2) three minutes unloaded warm-up, (3) 9.6±1.4 minutes ramp protocol (with 20.7±7.1 watts per minute increment), and (4) six minutes unloaded cycling for active recovery. All volunteers were encouraged to keep a constant cycling of 60 to 65 rpm and were stimulated to continue the CPET until volitional fatigue. The oxygen uptake and minute ventilation were measured breath-by-breath by a metabolic system (Vmax29c, Sensor Medics, Yorba Linda, CA, USA) calibrated before each experiment, according to the manufacturer's manual. Moreover, heart rate (HR) was calculated during the exercise based on a single lead ECG system (BioAmp FE132, ADInstruments, Australia).

The interruption criteria were according to previous work [16], and just one volunteer had the CPET interrupted due to the unexpected (excessively high) increase in arterial pressure. One volunteer was also excluded from this study due to oxygen desaturation on resting immediately before the CPET.

## Step three

The last step took seven days, where the vital signals from the wearable sensors were collected during unsupervised ADL, in an unobtrusive way, where participants maintained their daily routine. The volunteers were instructed to use the smart shirt for seven days, at least for eight active hours per day, except during showering and water activities. The smart shirt has three embedded sensors, and the raw signals were used to obtain biological and environmental data by a previously validated [30] proprietary algorithm. The HR data was measured by an ECG system (one-lead ECG channel, frequency: 256 Hz and with 12 bits resolution), and an algorithm that filters and averages the HR over the last 16 heartbeats. The breathing rate (BR), minute ventilation (Ve), and tidal volume (Vt) were estimated by the thoracic and abdominal belts. The Vt variable was obtained by dividing Ve by BR (Vt = Ve/BR). The respiratory belts were based on inductance plethysmography (sampled at 128 Hz with 16 bits resolution), and BR, Ve, and Vt were averaged over the last seven respiration cycles. The environmental data from total hip acceleration (Acc), and walking cadence (Cad) were based on triaxial accelerometer signals located at the right side of the hip. It was collected at 64 Hz, with a 13 bits resolution and a range of 16 g (with 0.004 g of resolution step). All data were resampled at 1 Hz.

## Data analysis

During the CPET, the following metabolic variables were measured: oxygen uptake ($\dot{V}O_2$); carbon dioxide output ($\dot{V}CO_2$); and respiratory exchange ratio (RER, or $\dot{V}CO_2/\dot{V}O_2$). Data were pre-processed in MatLab routine where the data were interpolated at 1 Hz, and then, the metabolic data were synchronized with HR, which was also used as a secondary criterion to confirm $\dot{V}O_{2-max}$, as described in Table 1 [31]. For each variable, the maximum (including the $\dot{V}O_{2-max}$, $RER_{max}$, and $HR_{max}$) was considered as the average of the last 20 s of the exercise protocol, before the CPET interruption. The $\dot{V}O_{2-max}$ (as a surrogate for CF) was considered as the ground truth for the machine learning algorithms training that should predict the $\dot{V}O_{2-max}$ (*pred* $\dot{V}O_{2-max}$) based on the inputs from the wearables and volunteer´s personal information.

Another MatLab routine was used to process the unobtrusive longitudinal data from the smart shirt. Initially, the 1-Hz variables were downloaded from the Hexoskin's dashboard (please check the documentation at https://www.hexoskin.com/pages/hexoskin-connected-

**Table 1. Characteristics of volunteers, peak variables obtained during the cardiopulmonary exercise testing, and the mean response of the variables obtained by the wearable.**

| Characteristics of Subjects (n = 43) | |
|---|---|
| **Anthropometric** | |
| Sex (M/F) | 32/11 |
| Age (years) | 37.50(25.00–55.00) |
| Weight (kg) | 75.41±13.12 |
| Height (m) | 1.74±0.09 |
| BMI (kg/m$^2$) | 24.84±3.12 |
| **CPET Peak** | |
| $\dot{V}O_{2-max}$ (l/min) | 2.42±0.80 |
| RER$_{max}$ | 1.28±0.10 |
| HR$_{max}$ (bpm) | 169.27(155.71–183.50) |
| **Wearable** | |
| μBR (rpm) | 17.92(16.48–19.77) |
| μ$V_E$ (l/min) | 15.86±4.71 |
| μAcc (g) | 0.06±0.03 |
| μHR (bpm) | 84.09±8.57 |
| Cad (spm) | 5.61(3.47–8.38) |
| μVt (ml/min) | 896.80±251.03 |

M: male; F: female; BMI: body mass index; CPET: cardiopulmonary exercise testing; $\dot{V}O_2$-max: maximum oxygen uptake; RER$_{max}$: maximum respiratory exchange ratio; HR$_{max}$: maximum heart rate; BR: breathing rate; $V_E$: minute ventilation; Acc (g): total hip acceleration, by gravity; Cad (spm): walking cadence, by steps per minute; μ: mean response.

health-platform –as of August 2021). Each downloaded dataset (~7 days) was combined into a single dataset consisting of 65±13 hours; For HR, beats/min lower than 30 and higher than 220 were excluded. For respiratory measurements, BR values lower than 3 and higher than 79 were used as a reference to exclude data also from Ve and Vt variables, beyond BR. Finally, the average response for all variables (μHR, μBR, μVe, μVt, μAcc, μCad) was computed and used as the inputs for the machine learning algorithms.

## Framework

As described before, the MatLab scripts were used to calculate the biological signal-derived inputs and the output data ($\dot{V}O_{2-max}$). Beyond the inputs from wearables (i.e., μHR, μBR, μVe, μVt, μAcc, and μCad), the age, sex, weight, height, and body mass index (BMI, weight/height$^2$) of each volunteer were also used as inputs to estimate the *pred* $\dot{V}O_{2-max}$ by machine learning algorithms (further explained). These steps are illustrated in Fig 1.

## Machine learning algorithm

Support vector machine (SVM) comprises a set of supervised learning algorithms used for classification and regression analysis. Introduced by Cortes and Vapnik [32], SVM is one of the most robust and flexible machine learning algorithms that has been successfully applied to several different problems [32]. In short, SVM algorithms build a model by finding a hyperplane in an n-dimensional space in which data points could be distinctly classified. Differently from other linear regression methods, the SVM algorithm creates a safety boundary from both sides of the hyperplane (known as margins), which is paramount information for better

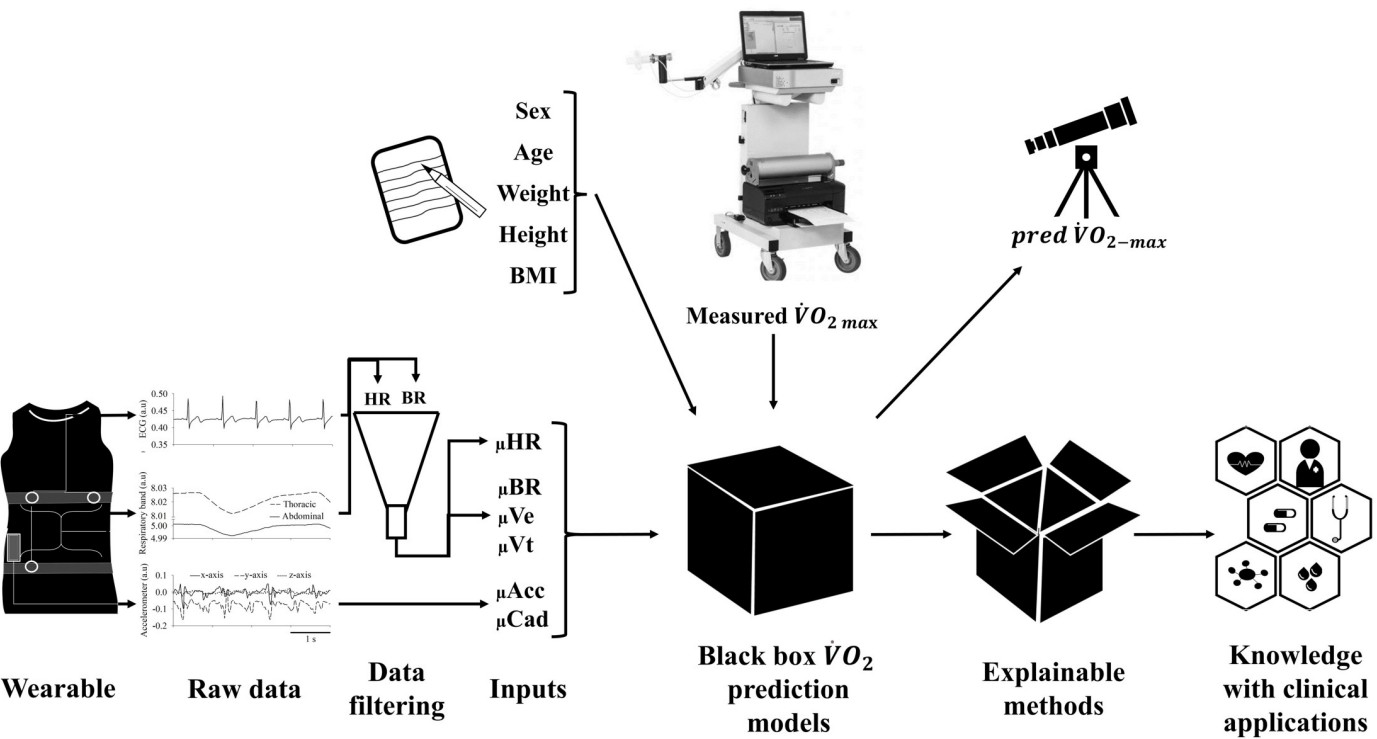

**Fig 1. The wearable system has embedded cardiac, respiratory, and movement sensors that measure unsupervised and unobtrusive biological data.** These raw data are processed, filtered, and averaged. Mean response to heart rate (μHR), breathing rate (μBR), minute ventilation (μVE), tidal volume (μVt), total hip acceleration (μAcc), and walking cadence (μCad), as well as sex, age, weight, height, and body mass index (BMI), were used as inputs to predict the maximum oxygen uptake ($pred\ \dot{V}O_{2-max}$). The resultant prediction model is a black box due to its high complexity and low explainability; therefore, explainable methods are necessary to extract meaningful knowledge that might have clinical applications.

modelling the uncertainties in the decision boundary zone considering two-class distribution. Thus, the SVM algorithm maximizes the separation (margin) between two classes in a higher-dimensional space from the input features. In the context of regression analysis, the SVM algorithm aims to find a linear function f(x) under the condition that f(x) is within a required accuracy epsilon from the y(x) of every data point, i.e., $|y(x)\text{-}f(x)| \leq \varepsilon$ where $\varepsilon$ is the distance between observed and predicted values for each data point. This work adopted the use of Support Vector Regression (SVR), an SVM formulation for regression problems [33] operating with a radial basis function as Kernel.

In turn, the SVM algorithms can produce more accurate and flexible models, despite the lack of some interpretability. SVM models can be considered partially interpretable models as we can determine which training data point is relevant for the prediction (i.e., the support vectors). On the other hand, it is hard to infer the contribution of features to the model's output as the input data points are projected into a higher-dimensional space to decide the predicted output value. To have accurate, and yet interpretable models, our methodology considers the use of a specific method for interpretable machine learning methods. In this work, we adopted the use of the SHapley Additive exPlanations (SHAP) method to estimate a local surrogate model, which was used to explain individual predictions. Thus, we can better manage the trade-off between accuracy and interpretability by taking advantage of robust regression algorithms and the SHAP method, which is described in the next section.

## Explainable methods

The ability to explain and interpret the prediction model's output is essential since the understanding of these models might also indicate how the human physiological systems interact with the environment. EXplainable Artificial Intelligence (XAI) is a growing research topic in the machine learning community and several methods have been proposed recently [34]. We can categorize the current approach for XAI as global and local explainable methods. While local methods provide explanations for each data point individually [35], global methods are able to provide explanations that make the entire model easier to understand, in addition to providing the rationale for the models to produce all possible results [34].

The SHapley Additive exPlanations (SHAP) approach [36] aims to explain the prediction of a given data point by computing the Shapley value for each feature input, which represents how much the features contribute to the model's prediction value. The concept of the Shapley value was originally introduced by Lloyd in the context of the cooperative game theory [37] that involves a fair distribution of both gains and costs to several players acting in coalition. Thus, Shapley value tries to ensure that each actor gains as much or more as they would have from acting independently. In explainable machine learning, the Shapley value of a feature input comprises its contribution to the model's prediction value, weighted and summed over all possible feature input combinations.

To evaluate the robustness and reliability of the regression models built in this study, we adopted the use of k-fold cross-validation. The evaluation protocol adopted in this study was designed to have predictions for each participant. To do this, we split the data into k folds (k = 9) disjoint among participants, which means that we do not have the same participant in two or more folds. Fig 2 illustrates the methodology adopted in this study. Typically, k-fold implementations available in well-known software packages fill the gaps by duplicating some arbitrary data points when there is no integer division between the data point and the number of folds. We decided to avoid this duplication due to the possibility of biasing our results. This strategy resulted in 9 values of R-squared, Mean Absolute Error (MAE) and Pearson correlation coefficient (R), and we assess the overall performance of our algorithm by computing the average of these metrics. Then, we estimated a regression model for each fold by using the k-th fold for validation purposes and the remaining folds for training purposes. To evaluate the effectiveness of built models, we computed the average of MAE and R between the observed ($\dot{V}O_{2-max}$) and predicted ($pred\ \dot{V}O_{2-max}$) value of $\dot{V}O_{2-max}$. For the MAE metrics low values are better, while for the Pearson correlation coefficient, values near 1.0 indicate a near-perfect correlation.

## Statistical analysis

We calculated the R and the Bland Altman plot to further investigate the agreement level between the $\dot{V}O_{2-max}$ and the $pred\ \dot{V}O_{2-max}$ for each volunteer. The Bland-Altman plot was done in Microsoft Excel (Office package 365, Microsoft Corporation, Redmond, WA), and the prediction quality was classified as "valid" when the R-value was higher than 0.7 [38].

The source of the data variability of the Shapley value of each input feature (i.e., their contribution for the predicted value of $pred\ \dot{V}O_{2-max}$) originated from the cross-validation method described above. The Shapley data normality was tested by the Shapiro Wilk test, and all the Shapley values presented a non-normal distribution. Thus, Friedman repeated-measures analysis along with the post-hoc Tukey Test was used to compare the final Shapley values (obtained from SHAP) between all inputs, since the Shapley values are normalized between the inputs.

In addition, the Shapley values for each eleven inputs were grouped into four domains: Anthropometric (age, weight, height, sex, and BMI), Hemodynamic (HR), Physical Activity

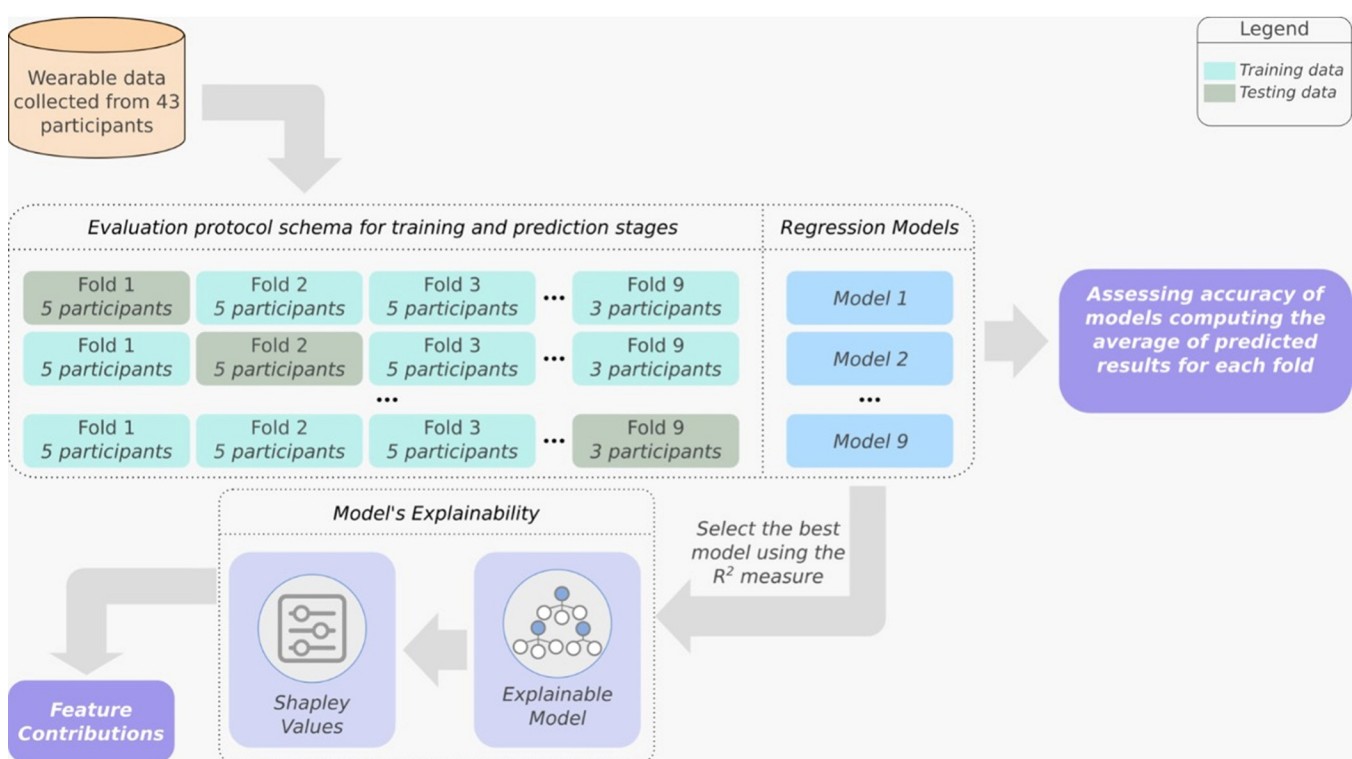

**Fig 2. Evaluation protocol adopted for this study.** Given a dataset containing wearable data from 43 participants, we first use the k-fold cross-validation (k = 9) to evaluate generalization aspects of regression models. Then, we use the R-squared measure to select the best model, which is used to build an explainable model via Shapley values to assess the feature contributions. We also computed the average Mean Absolute Error (MAE), and Pearson correlation to assess the accuracy of regression models.

(Acc and Cad), and Pulmonary (BR, Ve, and Vt). For each input, nine Shapley values were computed from the cross-validation. Afterward, within each domain, the Shapley values of the inputs were summed and divided by the number of inputs for this domain. Moreover, Friedman repeated-measures analysis or One Way Repeated Measures Analysis (depending on the data distribution tested by Shapiro-Wilk), with the post-hoc Tukey Test, were used to compare the final domain importance level between the four domains. Finally, the Spearman correlation (as the data were non-normally distributed) was used to verify the correlation level between all the Shapley values of the inputs (i.e., sex, age, weight, height, and BMI, BR, Ve, Acc, HR, Cad, and Vt).

Statistical analyses and graphs were done in Sigma Plot 14.0 (Systat Software Inc, Chicago, 2018). The statistical significance level (p) was set at 0.05.

## Results

As presented in Fig 3, 43 volunteers were included in the statistical analysis.

The demographic and anthropometric characteristics of the volunteers are presented in Table 1. From the 43 volunteers, 74.4% were men, and the age ranged from 19 to 72 years. According to a previous publication [39]; 5%, 21%, 51%, 21%, and 2% of the volunteers were classified as very low, low, fair, good, and excellent aerobic power, respectively accordingly their $\dot{V}O_{2-max}$ in relation to the participant's weight, which indicates a broad aerobic power spectrum. All CPET were interrupted due to volitional fatigue where the mean of maximum respiratory exchange ratio (RER$_{max}$) among all participants was higher than 1.1. In addition,

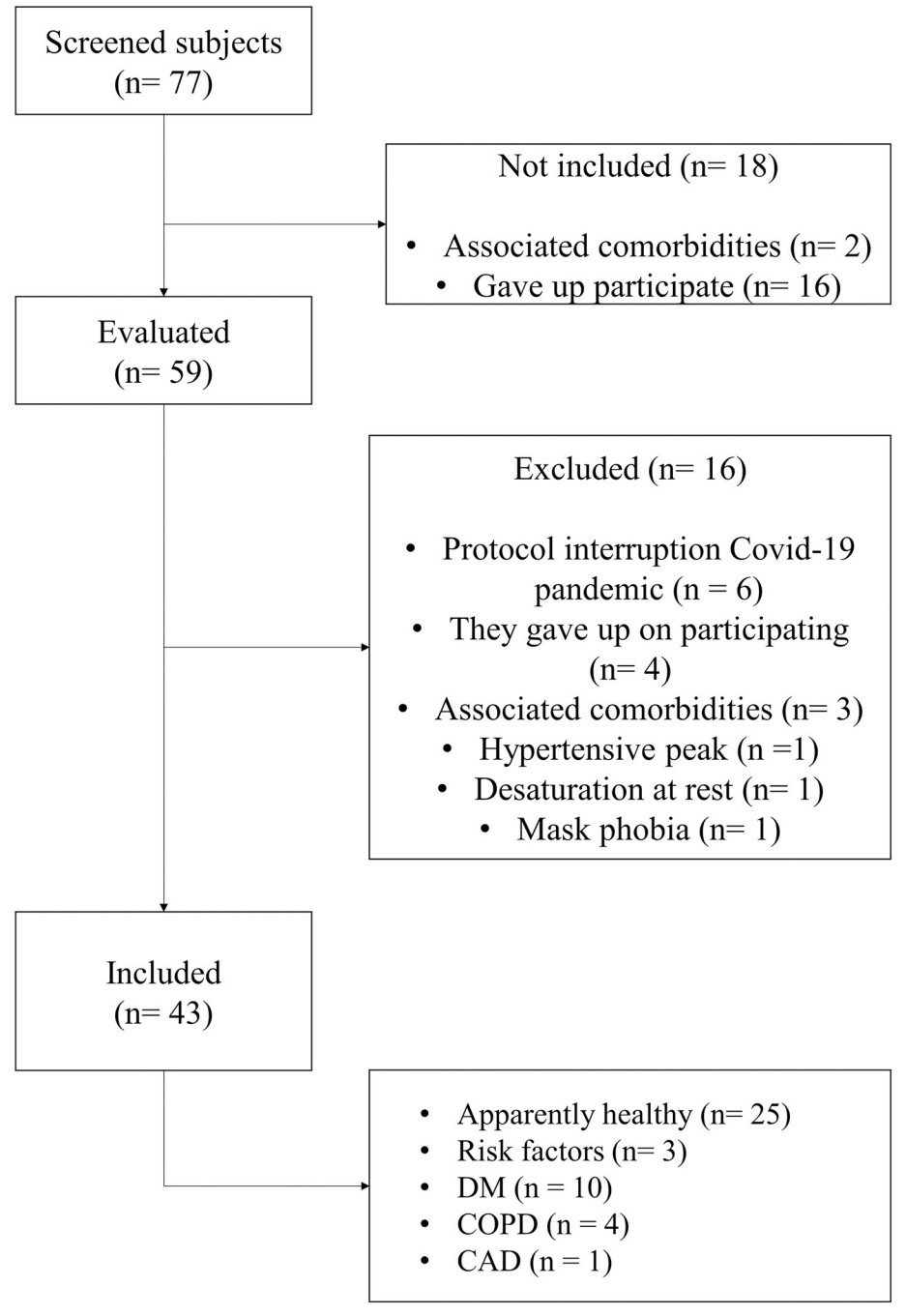

**Fig 3. Flowchart of screening, evaluation and inclusion and exclusion criteria for the study.** This flow diagram illustrates the sample size and the volunteer characteristics. DM: diabetes mellitus; COPD: chronic obstructive pulmonary disease; CAD: coronary artery disease.

the mean of the maximum heart rate (HR$_{max}$, described in Table 1) reached during CPET (among all participants) represented 91.5% of the predicted heart rate by age (HR$_{predicted}$ = 220-age) [31]. For each participant, the RER$_{max}$ and HR$_{max}$ were calculated as the mean value of the last 20 s of the incremental exercise. The number of days and time per day of the longitudinal data collection, as well as the average response for all variables ($\mu$BR, $\mu$Ve, $\mu$Vt, $\mu$HR, $\mu$Acc and $\mu$Cad) are also shown in Table 1.

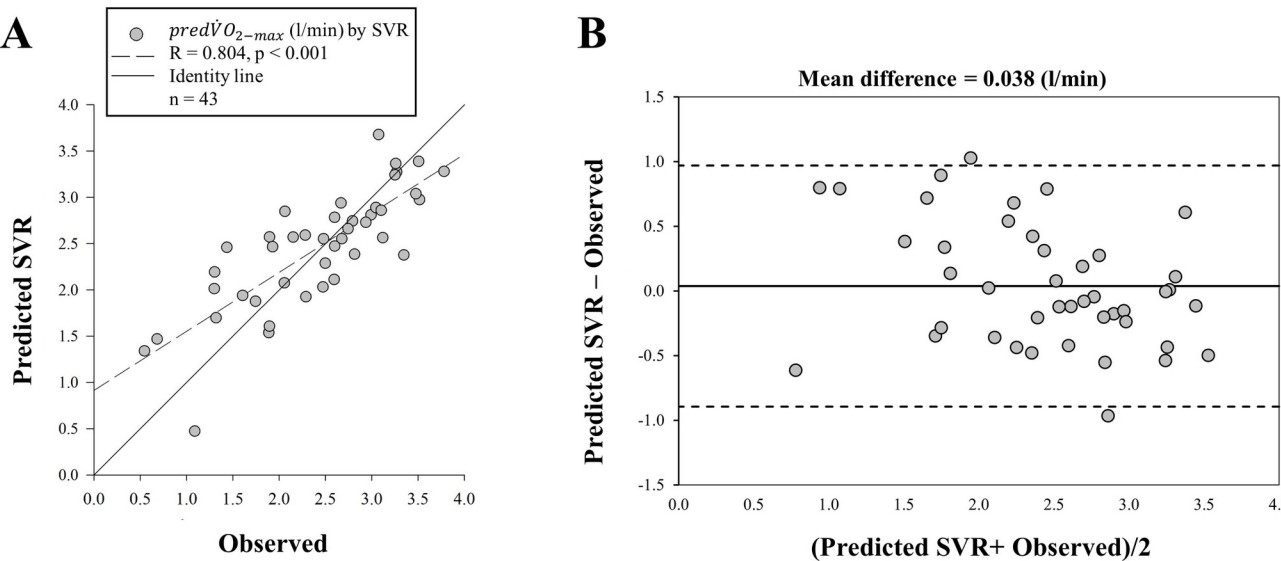

**Fig 4.** Linear correlation between maximum oxygen uptake during CPET and the predicted maximum oxygen uptake by machine learning technique (on letter A) and Bland-Altman plot of maximum oxygen uptake and prediction of the maximum oxygen uptake with the bias and the confidence interval ($CI_{95}$) (on letter B). Support vector regression (SVR); Pearson coefficient (R).

## General prediction model validation

The reproducibility and validity of the prediction of the $\dot{V}O_{2-max}$ ($pred\ \dot{V}O_{2-max}$) were tested using SVR, as described in Fig 4. The Bland-Altman analysis was used to verify the reproducibility between the $\dot{V}O_{2-max}$ measured during the CPET and the $pred\ \dot{V}O_{2-max}$. We found that the mean differences between model and observations was low (0.038 l/min). Furthermore, the agreement level by the Pearson correlation coefficient (R = 0.804, p< 0.001) was high and positive between the $\dot{V}O_{2-max}$ and the $pred\ \dot{V}O_{2-max}$.

## Evaluation protocol

As mentioned before, we generalization aspects of the built models by adopting the k-fold cross-validation evaluation protocol (see Fig 2). To measure the effectiveness of models, we computed the average of Mean Absolute Error (MAE) and the Pearson correlation coefficient (R) between the observed and predicted $\dot{V}O_{2-max}$ for each fold.

We observed a slight variability in the performance of the results achieved for the nine models. On average of MAE, the SVR regressors reached 0.384±0.134. In addition, the Pearson coefficient was high and positive (R = 0.8).

## Explainable models

According to the game theory [40], the resultant Shapley value from the above described SHAP method indicates the importance level of each input feature used to predict the variable $pred\ \dot{V}O_{2-max}$. Fig 5 shows the median Shapley values for each regression algorithm and the statistical differences between each input feature considered in this study. The feature age had the highest values, and the HR, height, weight, and Acc are ranked in the top-five list of the most important features. While the last four places are represented by respiratory measurements, such as the BR and Vt, Cad, and BMI. Moreover, when grouping the inputs into four domains (Anthropometric, Hemodynamic, Physical Activity, and Pulmonary), the

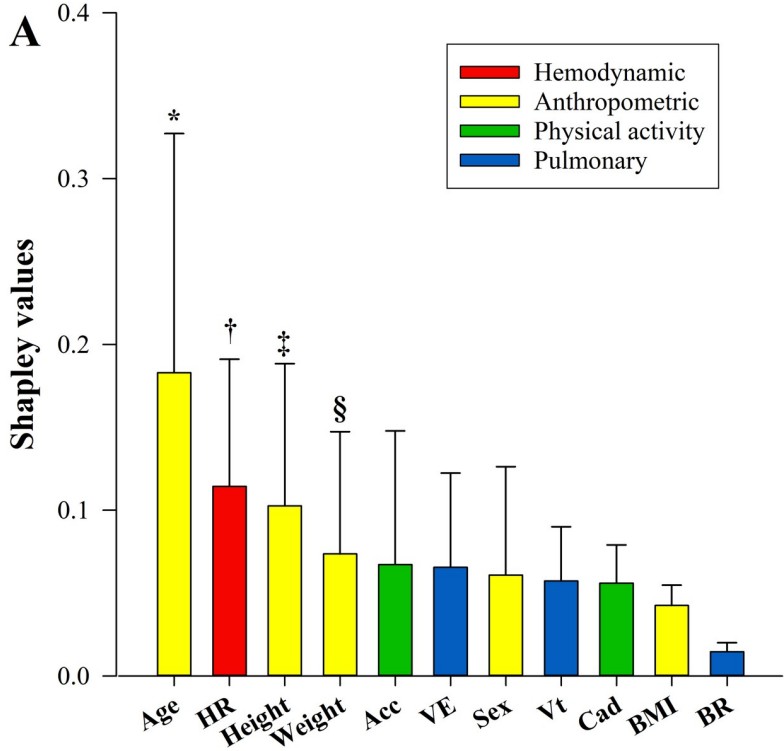

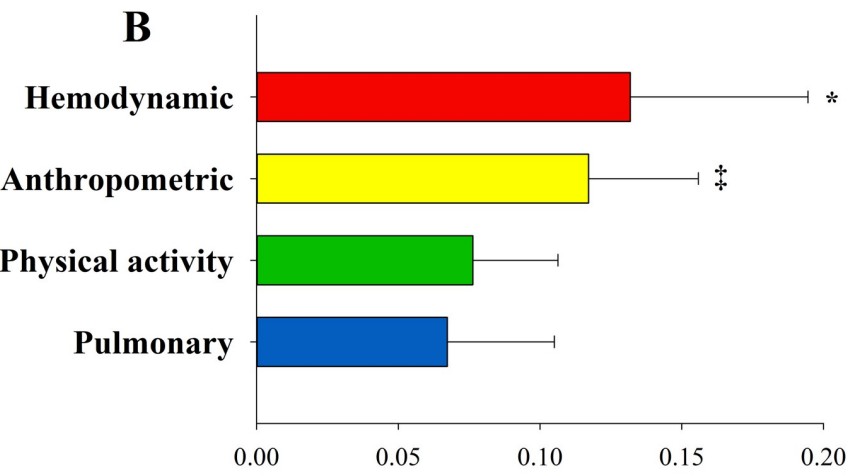

**Fig 5. Shapley values (importance level) of the inputs used to predict cardiovascular fitness. A**- Median and 25-75th percentile of Shapley values of the inputs from the Support Vector Regression (SVR). * Significant difference between age and BR (p > 0.001), between age and BMI (p > 0.001), between age and Cad (p = 0.006). † Significant difference between HR and BR (p = 0.003). ‡ Significant difference between height and BR (p = 0.004). § Significant difference between Weight and BR (p = 0.049). **B**- Mean±SD of Weighted average of Shapley values of the domains from the Support Vector Regression (SVR) model. * Significant difference between Hemodynamic and Physical Activity (p = 0.010), between Hemodynamic and Pulmonary (p = 0.003). ‡ Significant difference between Anthropometric and Pulmonary (p = 0.023). HR: heart rate, Acc: total hip acceleration; BMI: body mass index; BR: breathing rate, Vt: tidal volume; Cad: walking cadence, Ve: minute ventilation.

Hemodynamic domain presented statistically ($p<0.05$) higher importance to predict the $\dot{V}O_{2-max}$ compared with the Physical Activity and Pulmonary domains. Moreover, the Anthropometric domain was statistically ($p<0.05$) higher than the Pulmonary domain. We did not find any evidence of statistically significant differences ($p>0.05$) between the Hemodynamic and Anthropometric domains. Similarly, we also did not find any evidence of statistically significant differences between the Physical Activity and Pulmonary domains.

Finally, the correlations between the Shapley values of the inputs were calculated. We found some statistically ($p<0.05$) high and positive correlations between the Shapley values. For the SVR model, there were two high and positive correlations between Acc and age (R = 0.817, p = 0.004); and between minute ventilation (Ve) and height (R = 0.767, p = 0.012).

## Discussion

The SVR method showed to be reliable to predict maximal oxygen uptake (cardiovascular fitness), with an average MAE of 0.38±0.13 l/min. Afterward, we identified the most important inputs (and their respective domains) modelled to predict the $\dot{V}O_{2-max}$, using the explainable model. Hemodynamic and Anthropometric domains were more important to predict cardiovascular fitness than the Physical Activity and Pulmonary domains.

As previously reported, the CF measurement by CPET was predicted by machine learning techniques [24, 25], however the $\dot{V}O_{2-max}$ prediction from data exclusively obtained during unobtrusive ADL protocols is still under investigation. Our results corroborate with previous studies, as mentioned before as the *pred* $\dot{V}O_{2-max}$ and $\dot{V}O_{2-max}$ were statistically similar (p = 0.602 for SVR) and the predictions were reliable as verified by the low MAE and high R. Moreover, our mean errors of the Bland-Altman were low (SVR = 0.038 l/min), and these results are very close to what were previously described in a study (0.22 l/min) [41] that used the same wearables that we used. However, our agreement limits were higher (0.970 to -0.894 l/min) than Amelard, Hedge and Hughson, 2021 (0.218 to—0.262 l/min), although our data being within the agreement limits.

The SVR using a radial basis function (RBF) as Kernel, can estimate a non-linear machine learning model that takes a small number of critical boundary samples, called support vectors. These optimize the predictions compared with models limited to single-dimension linear boundaries, such as the linear regression [42]. Therefore, SVR should allow the expression of more complexities from the input-output relationships, improving the results of regressions. In addition, SVR has been used in Medical Sciences to predict coronary artery disease and stroke [19].

Similar to our study, previous reports [24] used SVR models to identify the activity levels of unsupervised ADL from HR and accelerometer data in healthy adults (both sexes, 25.1±6.0 years, 22.7±2.5 kg/m$^2$). These authors used linear regression models to estimate the CF by wearables. In our study, we used unobtrusive longitudinal data collected by a wearable system that also considered more physiological inputs (such as the Ve and BR from the respiratory sensors). These signals were then used to train machine learning models to predict the $\dot{V}O_{2-max}$ of volunteers with a broad spectrum of aerobic power, which included (contrary to previous publications) apparently healthy volunteers with risk factors for NCDs and with NCDs. Therefore, our results add to the current literature to further support the use of machine learning models to predict CF in the general population, including diseased groups.

Once the prediction models were validated for this broad spectrum of CF, the Shapley method was applied to check the importance level of these 11 measured inputs (i.e., sex, age, height, weight, BMI, µBR, µVe, µAcc, µHR, µCad, and µVt). All inputs were considered as

"good players", according to the Shapley value that represents the contribution of each input feature to predict CF [28, 43].

It is worth mentioning that the Shapley values are able to better isolate each input individual influence over the predictions, thus it is less influenced by expected multicollinearities between the input features [43, 44], as we expect in the relationship between age and CF, for example. The values were expressed as positive (all the cases in our study) or negative, which means that the contribution of a particular input led to a better output prediction, i.e., *pred* $\dot{V}O_{2-max}$ [45].

Explainable methods are used as an explanatory tool for complex models, especially when the resultant model is derived from a machine learning approach with a considerable number of hyperparameters and weights or coefficients, such as the SVR method. Thus, the Shapley values provided an approximation for the global input importance in predictions of complex responses, as we expect from biological systems [28, 46]. In fact, the CF level depends on several factors [47–51] that can be tracked by some inputs used in our study, including those measured by the wearable sensors.

Among all the inputs, age was the most important variable for both models, which also corroborates with the model-agnostic characteristics of the SHAP method. In a systematic review that aimed to identify the determinants of CF, the authors found that more than 80% of the studies identified an inverse relationship between CF and age [51]. This relationship might be justified by the influences of the aging process over the aerobic response, including the reduction of lean mass. In addition, the decrease of the $\dot{V}O_{2-max}$ by aging, might be also related to the reduction of the activity levels and the diseased states [52, 53].

When comparing the Shapley values (importance levels), the feature age was highly (and positively) correlated with the Shapley value of feature Acc. Thus, when age was more important for the CF definition (i.e., prediction), Acc was also more important. In a longitudinal follow-up of 8 years, Katzel, Sorkin, and Fleg 2001, found that maintaining a high level of training is inversely related to the rate of decline of aerobic power, due to aging [54]. Thus, the SVR model was able to identify this expected observation. It is known that the $\dot{V}O_{2-max}$ during CPET on cycle ergometry is influenced by sex, age, weight, and height [55]. Thus, in our study, the Anthropometric domain of the inputs took second place for the CF prediction. In addition, this domain was not statistically different from the Hemodynamic domain, but statistically different from the Pulmonary domain.

Between the vital signals measured by the wearable system, the Hemodynamic domain (evaluated here by the μHR) was the most important input for the $\dot{V}O_{2-max}$ prediction, which, to some extent, corroborates with previous literature [51] that associated resting HR with maximal aerobic power. High values of resting HR were related to low values of $\dot{V}O_2$ and consequently, lower levels of CF [56]. Altini et al., 2016 [25] found that HR explained 64% of CF variability when including sex, weight, and age as predictors, and this percentage rise as the intensity of physical activities increased. In our results, the μHR was very crucial to our predictions, maybe because μHR also includes a great deal of information regarding the resting HR as it was calculated as the average HR response throughout 7 days that comprise many resting periods.

## Study limitations

Some limitations of the present study should be considered. As described before, the feature extraction method (simple average of the longitudinal signal) might have reduced the complexity of the data from the wearable system. Thus, more studies are necessary to develop new feature extraction methods for mining longitudinal data and extracting more complex and meaningful information. Although the volunteers wore the wearables for most of the day, they

did not use it full time, including during sleep, and two volunteers practiced swimming as their main sport activity. Collecting information during sleep and water activities might improve the understanding of the CF from longitudinal data obtained from wearable sensors. It is known that HR and mean blood pressure have been used for assessing hemodynamic conditions, however assessing blood pressure must be done invasively or indirectly (which depends on the HR to be calculated), [57] thus we only consider HR as the hemodynamic domain. In this study, we used the SVR with an RFB Kernel, which means that we have a two-dimensional parameter space. Due to the low complexity of searching hyperparameters in this parameter space, we adopted a simple, effective, and well-known method named Grid Search [58]. In short, given a set of values for the variables that comprise the model, i.e., the parameters C and Gamma, the Grid Search algorithm makes a direct search on a set of all trials, which is formed by assembling every possible combination of values for the C and Gamma. This approach does not guarantee the best values for the parameters, which could be a limitation of this study. However, it is one of the most widely used approaches for hyperparameter training in machine learning [58].

## Conclusion

Cardiovascular fitness can be predicted by wearable technologies associated with artificial intelligence. Explainable models can be used to extract clinical insights from these predictions. Therefore, the $pred\ \dot{V}O_{2-max}$ showed to be reproducible and valid in volunteers apparently healthy, with risk factors to develop NCDs and with NCDs. Thus, the association between longitudinal and unobtrusive biological data from wearables, machine learning, and explainable models represents a unique framework in Health Science.

## Supporting information

**S1 Dataset.**
(XLSX)

## Author Contributions

**Conceptualization:** Maria Cecília Moraes Frade, Thomas Beltrame, Mariana de Oliveira Gois, Aparecida Maria Catai.

**Data curation:** Maria Cecília Moraes Frade, Thomas Beltrame, Mariana de Oliveira Gois, Allan Pinto, Silvia Cristina Garcia de Moura Tonello, Ricardo da Silva Torres, Aparecida Maria Catai.

**Formal analysis:** Maria Cecília Moraes Frade, Thomas Beltrame, Mariana de Oliveira Gois, Allan Pinto, Silvia Cristina Garcia de Moura Tonello, Ricardo da Silva Torres, Aparecida Maria Catai.

**Funding acquisition:** Maria Cecília Moraes Frade, Thomas Beltrame, Mariana de Oliveira Gois, Ricardo da Silva Torres, Aparecida Maria Catai.

**Investigation:** Maria Cecília Moraes Frade, Thomas Beltrame, Mariana de Oliveira Gois, Allan Pinto, Silvia Cristina Garcia de Moura Tonello, Ricardo da Silva Torres, Aparecida Maria Catai.

**Methodology:** Maria Cecília Moraes Frade, Thomas Beltrame, Mariana de Oliveira Gois, Allan Pinto, Ricardo da Silva Torres, Aparecida Maria Catai.

**Project administration:** Maria Cecília Moraes Frade, Thomas Beltrame, Mariana de Oliveira Gois, Aparecida Maria Catai.

**Resources:** Maria Cecília Moraes Frade, Thomas Beltrame, Mariana de Oliveira Gois, Ricardo da Silva Torres, Aparecida Maria Catai.

**Supervision:** Thomas Beltrame, Mariana de Oliveira Gois, Ricardo da Silva Torres, Aparecida Maria Catai.

**Validation:** Maria Cecília Moraes Frade, Thomas Beltrame, Mariana de Oliveira Gois, Aparecida Maria Catai.

**Visualization:** Maria Cecília Moraes Frade, Thomas Beltrame, Mariana de Oliveira Gois, Allan Pinto, Silvia Cristina Garcia de Moura Tonello, Ricardo da Silva Torres, Aparecida Maria Catai.

**Writing – original draft:** Maria Cecília Moraes Frade, Thomas Beltrame, Mariana de Oliveira Gois, Allan Pinto, Silvia Cristina Garcia de Moura Tonello, Ricardo da Silva Torres, Aparecida Maria Catai.

**Writing – review & editing:** Maria Cecília Moraes Frade, Thomas Beltrame, Mariana de Oliveira Gois, Allan Pinto, Silvia Cristina Garcia de Moura Tonello, Ricardo da Silva Torres, Aparecida Maria Catai.

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
