## [Decision Letter · Decision Letter 0]

8 Aug 2022

PONE-D-22-13469Toward Characterizing Cardiovascular Health using Machine Learning based on Unobtrusive dataPLOS ONE

Dear Dr. Beltrame,

Thank you for submitting your manuscript to PLOS ONE. After careful consideration, we feel that it has merit but does not fully meet PLOS ONE’s publication criteria as it currently stands. Therefore, we invite you to submit a revised version of the manuscript that addresses the points raised during the review process.

ACADEMIC EDITOR:  Dear Author, Please revise your manuscript and make the necessary corrections as suggested by the reviewers. The decision of this manuscript is justified based on PLOS ONE’s publication criteria and not on its novelty or perceived impact.

We look forward to receiving your revised manuscript.

Kind regards,

Zulkarnain Jaafar

Academic Editor

PLOS ONE

Journal Requirements:

Reviewers' comments:

Reviewer's Responses to Questions

**Comments to the Author**

1. Is the manuscript technically sound, and do the data support the conclusions?

Reviewer #1: Yes

Reviewer #2: Partly

2. Has the statistical analysis been performed appropriately and rigorously? 

Reviewer #1: Yes

Reviewer #2: Yes

3. Have the authors made all data underlying the findings in their manuscript fully available?

Reviewer #1: Yes

Reviewer #2: Yes

4. Is the manuscript presented in an intelligible fashion and written in standard English?

Reviewer #1: Yes

Reviewer #2: Yes

5. Review Comments to the Author

Reviewer #1: Great work.

The use of digital technologies and Artificial intelligence is the growing world of knowledge and the making use of them through research adds another milestone to the advancement of the digital technologies in preventing future illness, which is by far more important in disease prevention.

Reviewer #2: Line 49: the reference needs to be updated to reflect NCDs risk factors and not specifically CVD.

Line 50 – 51: The authors seem to be including CVD as part of NCDs (paragraph 1) and now talking about NCDs impact on CVD. Please review.

Line 50: Not all NCDs are directly impacted by Cardiovascular health. The paper would benefit from clearly delineating CVD focus on not all NCDs.

Line 50: Suggest rewording cardiovascular health (CH) to cardiovascular fitness (CF). CH is very broad within which CF is focused area.

Line 54-56: The use of CPET in assessing cardiovascular health is one metrics for CVD risk. Only 1 of the three papers actually reference cardiorespiratory fitness. Also for subclinical CVD risk, CPET is not a scalable tool as there are other measures more closely aligned to overall risk.

Line 76: “pre-symptomatic detection of NCDs”. Please revise and focus on cardiovascular diseases.

6. PLOS authors have the option to publish the peer review history of their article (what does this mean?). If published, this will include your full peer review and any attached files.

Reviewer #1: No

Reviewer #2: No

---

## [Author Response · Author response to Decision Letter 0]

13 Oct 2022

Response to the Reviewer:

We appreciate all the comments from the reviewers. Improvements throughout the manuscript were made in accordance with the reviewers’ comments. The comments were extremely valuable for improving the quality of our manuscript. Please find below our responses and comments related to all the reviewers’ concerns.

Reviewer 1: Great work.

The use of digital technologies and Artificial intelligence is the growing world of knowledge and the making use of them through research adds another milestone to the advancement of the digital technologies in preventing future illness, which is by far more important in disease prevention.

Response: Thank you very much for your comment. We are so grateful to have such a positive comment.

Reviewer 2: 

Line 49: the reference needs to be updated to reflect NCDs risk factors and not specifically CVD.

Response: Thank you for your comment. We agree with you, thus we added the following citation “Budreviciute A, Damiati S, Sabir DK, Onder K, Schuller-Goetzburg P, Plakys G, Katileviciute A, Khoja S and Kodzius R (2020) Management and Prevention Strategies for Non-communicable Diseases (NCDs) and Their Risk Factors. Front. Public Health 8:574111. doi: 10.3389/fpubh.2020.574111”.

Line 50 – 51: The authors seem to be including CVD as part of NCDs (paragraph 1) and now talking about NCDs impact on CVD. Please review.

Response: Thank you for your comment, we rewrote the statement following your comment. 

Line 50: Not all NCDs are directly impacted by Cardiovascular health. The paper would benefit from clearly delineating CVD focus on not all NCDs.

Line 50: Suggest rewording cardiovascular health (CH) to cardiovascular fitness (CF). CH is very broad within which CF is focused area.

Response: Thank you for your comments. We corrected this sentence as you suggested. The term “cardiovascular healthy (CH)” was replaced by “Cardiovascular fitness (CF)”.

Line 54-56: The use of CPET in assessing cardiovascular health is one metrics for CVD risk. Only 1 of the three papers actually reference cardiorespiratory fitness. Also for subclinical CVD risk, CPET is not a scalable tool as there are other measures more closely aligned to overall risk.

Response: We would like to thank the reviewer for raising this issue. Although the term “Cardiovascular health and cardiovascular fitness” are used interchangeably sometimes we agree with the reviewer, so the manuscript was changed accordingly. Regarding the CPET application, we also added a sentence in line 64 to fit your suggestion. 

Line 76: “pre-symptomatic detection of NCDs”. Please revise and focus on cardiovascular diseases.

Response: Thank you for your comment, according to your suggestion we added more information (in line 80) to the manuscript.

Sincerely,

The Authors

---

## [Decision Letter · Decision Letter 1]

4 Dec 2022

PONE-D-22-13469R1Toward Characterizing Cardiovascular Fitness Using Machine Learning Based on Unobtrusive DataPLOS ONE

Dear Dr. Beltrame,

Thank you for submitting your manuscript to PLOS ONE. After careful consideration, we feel that it has merit but does not fully meet PLOS ONE’s publication criteria as it currently stands. Therefore, we invite you to submit a revised version of the manuscript that addresses the points raised during the review process.

ACADEMIC EDITOR: Dear Author, Please attend to all of the reviewers' comment and make the necessary corrections as suggested. The decision of this manuscript is justified based on PLOS ONE’s publication criteria and not on its novelty or perceived impact.

We look forward to receiving your revised manuscript.

Kind regards,

Zulkarnain Jaafar

Academic Editor

PLOS ONE

Reviewers' comments:

Reviewer's Responses to Questions

**Comments to the Author**

1. If the authors have adequately addressed your comments raised in a previous round of review and you feel that this manuscript is now acceptable for publication, you may indicate that here to bypass the “Comments to the Author” section, enter your conflict of interest statement in the “Confidential to Editor” section, and submit your "Accept" recommendation.

Reviewer #1: All comments have been addressed

Reviewer #3: All comments have been addressed

2. Is the manuscript technically sound, and do the data support the conclusions?

Reviewer #1: Yes

Reviewer #3: Yes

3. Has the statistical analysis been performed appropriately and rigorously? 

Reviewer #1: Yes

Reviewer #3: No

4. Have the authors made all data underlying the findings in their manuscript fully available?

Reviewer #1: Yes

Reviewer #3: Yes

5. Is the manuscript presented in an intelligible fashion and written in standard English?

Reviewer #1: Yes

Reviewer #3: Yes

6. Review Comments to the Author

Reviewer #1: I have no specific comment to the authors.

All my comments and suggestions are well addressed and the authors did an incredible work.

Reviewer #3: I read the manuscript interestingly and I found it so informative. Overall, the topic is interesting and important. Although this manuscript is well written and on a good subject, some issues need to be addressed:

Line 38: SVR stands for what? Please writhe the full form of abbreviation in the first appearance.

Line 37-40: Please re-write this section and provide separate results and conclusions.

Introduction is too long and boring.

The significance level is 0.05 not 5%.

Only 43 cases were included in the analysis. The sample size of 43 is too small to causal analysis and adjusting different variables. Explain it or provide a valid formula?

What was your criterion to select the variable to include in the models? The best recommended method is using directed acyclic graph (DAG) which helps you to find confounders, colliders and mediators.

Please provide a list of algorithms used in the SVM.

7. PLOS authors have the option to publish the peer review history of their article (what does this mean?). If published, this will include your full peer review and any attached files.

Reviewer #1: **Yes: **Genemo, Gebi Agero

Reviewer #3: No

---

## [Author Response · Author response to Decision Letter 1]

31 Jan 2023

Dear Editor,

We would like to thank you for allowing us to resubmit our manuscript. The modifications have been highlighted throughout the manuscript and their position in the manuscript has been reported in the response to referees. In addition, we also updated the values in table 1 after we double-checked our dataset which is now being shared in our submission. These new values did not change the statistical results as well as our study outcome.

We hope that now it is suitable for publication in PlosOne.

Response to the Reviewer:

We appreciate all the comments from the reviewers. Improvements throughout the manuscript were made in accordance with the reviewer comments. The comments were extremely valuable for improving the quality of our manuscript. Please find below our responses and comments related to all the reviewer concerns.

Reviewer 1: I have no specific comment to the authors.

All my comments and suggestions are well addressed and the authors did an incredible work.

Response: Thank you very much.

Reviewer 3: I read the manuscript interestingly and I found it so informative. Overall, the topic is interesting and important. Although this manuscript is well written and on a good subject, some issues need to be addressed:

Response: Thank you very much for this opportunity to improve our manuscript. We corrected all points that you mentioned.

Line 38: SVR stands for what? Please writhe the full form of abbreviation in the first appearance.

Response: Thank you for your comment, we rewrote the statement.

Line 37-40: Please re-write this section and provide separate results and conclusions.

Response: Thank you for your comment. We added the conclusion: “Therefore, we conclude that the cardiovascular fitness can be predicted by wearable technologies associated with machine learning during unsupervised activities of daily living. ”

Introduction is too long and boring.

Response: Thank you for your comment. We deleted some sentences in the introduction to make this section shorter and easier to read. 

The significance level is 0.05 not 5%.

Response: Thank you for your comment, we corrected the statement.

Only 43 cases were included in the analysis. The sample size of 43 is too small to causal analysis and adjusting different variables. Explain it or provide a valid formula?

Response: We would like to thank the reviewer for raising this issue. Following your comment, we calculated the power of the sample a posteriori, considering as the main outcome the measured and predicted the V ˙O_(2-max), as well as a correlation coefficient of 0.804 and a p value of 0.05. The test results are reported below (GPower 3.1 software):

Exact - Correlation: Bivariate normal model

Options: exact distribution

Analysis: Post hoc: Compute achieved power 

Input: Tail(s) = Two

 Correlation ρ H1 = 0.804

 α err prob = 0.05

 Total sample size = 43

 Correlation ρ H0 = 0

Output: Lower critical r = -0.3007930

 Upper critical r = 0.3007930

 Power (1-β err prob) = 0.9999996

Thus, we found that for our sample size of 43 individuals, with an observed R of 0.804 (V ˙O_(2-max) and the predV ˙O_(2-max)) and a p value of 0.05, the statistical power was 99.9%, which can be considered as a strong statistical power. As our SVR algorithm was successfully trained, we believe that its inputs carry information that explains the output from the model (knowledge extraction).

What was your criterion to select the variable to include in the models? The best recommended method is using directed acyclic graph (DAG) which helps you to find confounders, colliders and mediators.

Response: We thank the reviewer for this observation. We assume that the reviewer refers to the use of DAG for the hyperparameter selection process.

In this study, we used the SVR with an RFB Kernel, which means that we have a two-dimensional parameter space. Due to the low complexity of searching hyperparameters in this parameter space, we adopted a simple, effective, and well-known method named Grid Search [1]. In short, given a set of values for the variables that comprise the model, i.e., the parameters C and Gamma, the Grid Search algorithm makes a direct search on a set of all trials, which is formed by assembling every possible combination of values for the C and Gamma. We agree with the reviewer that this approach does not guarantee the best values for the parameters [1], and thus better approaches for doing that, as pointed out by the reviewer, can potentially give us better results, we would like to keep this hyperparameter search simple due mainly because of the number of samples that we have available for training. Recall that, in several applications, the use of Grid Search often leads to the choice of hyperparameter values that are effective in practice. Therefore, it is one of the widely used approaches for hyperparameter definition in machine learning tasks. 

In order to attend to the comment of the reviewer, we added a paragraph in the manuscript to inform the reader about the mentioned limitation and the suggested methodology for future improvement of this work.

[1] Bergstra, James, and Yoshua Bengio. "Random search for hyper-parameter optimization." Journal of machine learning research 13.2 (2012).

Please provide a list of algorithms used in the SVM.

Response: Thank you for your comment. Please follow the list of algorithms used in the SVM.

- R.-E. Fan, K.-W. Chang, C.-J. Hsieh, X.-R. Wang, and C.-J. Lin. LIBLINEAR: A library for large linear classification Journal of Machine Learning Research 9(2008), 1871-1874.

- C.-C. Chang and C.-J. Lin. LIBSVM: a library for support vector machines. ACM Transactions on Intelligent Systems and Technology, 2:27:1--27:27, 2011.

Sincerely,

The Authors

---

## [Decision Letter · Decision Letter 2]

15 Feb 2023

Toward Characterizing Cardiovascular Fitness Using Machine Learning Based on Unobtrusive Data

PONE-D-22-13469R2

Dear Dr. Beltrame,

We’re pleased to inform you that your manuscript has been judged scientifically suitable for publication and will be formally accepted for publication once it meets all outstanding technical requirements.

Kind regards,

Zulkarnain Jaafar

Academic Editor

PLOS ONE

Additional Editor Comments (optional):

Reviewers' comments:

Reviewer's Responses to Questions

**Comments to the Author**

1. If the authors have adequately addressed your comments raised in a previous round of review and you feel that this manuscript is now acceptable for publication, you may indicate that here to bypass the “Comments to the Author” section, enter your conflict of interest statement in the “Confidential to Editor” section, and submit your "Accept" recommendation.

Reviewer #1: All comments have been addressed

Reviewer #3: All comments have been addressed

2. Is the manuscript technically sound, and do the data support the conclusions?

Reviewer #1: Yes

Reviewer #3: Yes

3. Has the statistical analysis been performed appropriately and rigorously? 

Reviewer #1: Yes

Reviewer #3: Yes

4. Have the authors made all data underlying the findings in their manuscript fully available?

Reviewer #1: Yes

Reviewer #3: Yes

5. Is the manuscript presented in an intelligible fashion and written in standard English?

Reviewer #1: Yes

Reviewer #3: Yes

6. Review Comments to the Author

Reviewer #1: All the comments are well addressed.

The manuscript is written in clear, correct, and unambiguous language.

Reviewer #3: I check the revised manuscript interestingly. I have no more comment to the authors. All comments are addressed.

7. PLOS authors have the option to publish the peer review history of their article (what does this mean?). If published, this will include your full peer review and any attached files.

Reviewer #1: No

Reviewer #3: No

---

## [Editor Report · Acceptance letter]

22 Feb 2023

PONE-D-22-13469R2 

Toward Characterizing Cardiovascular Fitness using Machine Learning based on Unobtrusive data 

Dear Dr. Beltrame:

I'm pleased to inform you that your manuscript has been deemed suitable for publication in PLOS ONE. Congratulations! Your manuscript is now with our production department. 

Kind regards, 

on behalf of

Dr. Zulkarnain Jaafar 

Academic Editor

PLOS ONE